# Exploiting Opportunistic Scheduling Schemes and WPT-Based Multi-Hop Transmissions to Improve Physical Layer Security in Wireless Sensor Networks [note 1]

**DOI:** 10.3390/s19245456

**Published:** 2019-12-11

**Authors:** Kyusung Shim, Toan-Van Nguyen, Beongku An

**Affiliations:** 1Department of Electronics and Computer Engineering, Hongik University, Sejong City 30016, Korea; shimkyusung@outlook.kr (K.S.); vannguyentoan@gmail.com (T.-V.N.); 2Department of Software and Communications Engineering, Hongik University, Sejong City 30016, Korea

**Keywords:** artificial noise, multi-hop transmission, opportunistic scheduling, physical layer security, secrecy outage probability, wireless sensor networks

## Abstract

This paper studies the secrecy performance of wireless power transfer (WPT)-based multi-hop transmissions in wireless sensors networks (WSNs), where legitimate nodes harvest energy from multiple power beacons (PBs) to support the multi-hop secure data transmission to a destination in the presence of an eavesdropper. Specifically, the PBs not only transfer radio frequency energy to the legitimate nodes but also act as friendly jammers to protect data transmission. To improve secrecy performance, we propose two secure scheduling schemes, named minimum node selection (MNS) scheme and optimal node selection (ONS) scheme. We then evaluate the performance of the proposed schemes in terms of the exact closed-form for secrecy outage probability (SOP) and asymptotic SOP. The developed analyses are verified by Monte-Carlo simulations. The numerical results show that the ONS scheme outperforms the MNS scheme emerging as an effective protocol for secure multi-hop transmission in WSNs. Furthermore, the effects of the number of PBs, number of hops, time switching ratio, and the secure target data rate on the system performance are also investigated.

## 1. Introduction

Wireless sensor networks (WSNs) has been emerged as a networking solution for future wireless networks supporting Internet-of-thing (IoT) ecosystems [1,2]. In practice, WSNs are deployed in the battlefield or hazardous environments to allow collecting and monitoring the physical phenomena without human interaction [3,4]. However, one of the most challenges faced by traditional WSNs is how to maintain the network lifetime since the sensor nodes are low cost, small size, and have limited power supply [5]. Moreover, energy for encoding and protecting information against eavesdropping is also raising a significant burden on the traditional WSNs and IoT systems. Wireless power transfer (WPT) is a sustainable energy solution to efficiently solve such a problem in WSNs since each sensor node can harvest energy from multiple power beacons (PBs) to maintain their operation [6,7,8]. Specifically, the authors in [9] introduced the time-switching-based relaying (TSR) and power splitting-based relaying (PSR) architectures to allow energy harvesting and information processing at a battery-less relay node, which enables a wide range of attractive applications with stringent quality of services in future IoT systems and WSNs. Recently, the researches on energy harvesting address the harvesting energy from the light-emitting diode (LED) as well as the radio frequency (RF) in indoor IoT systems [10].

Multi-hop transmission has been identified as an effective technique to extend the networks coverage in WSNs by leveraging information transmission from a source node to a destination node via multiple intermediate sensor nodes. However, the broadcast of the wireless signals in multi-hop transmissions is easily wiretapped by illegitimate users [11]. Physical layer security (PLS) is one of the most effective approaches to protect the information against eavesdroppers by exploiting physical characteristics of wireless channels [12]. Moreover, the PLS concept can be naturally applied to the WPT-based multi-hop transmission in WSNs, where the PBs not only radiate RF signals to power to the sensor nodes but also generate artificial noise to degrade the eavesdropping channels. The secrecy outage probability (SOP) is one of the system metrics to evaluate secure performance, and is defined as the probability that the difference between capacity of the main channel and that of the eavesdropper channel falls below a predefined secrecy target data rate [13,14]. Additionally, perfect secure transmission is achieved when the channel state information (CSI) of the main channel is higher than that of the eavesdropper channel [15].

### 1.1. Related Works and Motivations

In this subsection, we discuss the most recent works for the PLS and WPT-based enabling multi-hop transmission in WSNs. Do et al. in [16] proposed the relay selection schemes to enhance the system performance under independent and non-identical distribution (i.n.i.d.) fading channel. The paper also proposed the PSR mechanism for relays to harvest energy from the source RF radiation. The authors in [17] proposed two antenna selection schemes by using the CSI to improve secrecy performance in underlay cognitive radio network. Additionally, the secondary transmitter can harvest energy from the primary transmitter’s signal to transmit the message to a secondary receiver. In [18], the authors considered the cooperative relaying system, where the relay is equipped with TSR architecture to support data transmission from a source and a destination with maximal ratio combining (MRC) technique. In [19], the authors considered the wireless energy harvesting (EH) multi-hop cluster-based networks, where the destination in the considered networks is equipped with multiple antenna and proposed relay selection schemes to enhance the system performance. In our previous work [20], we proposed the WPT-based multi-hop transmission model in the presence of an eavesdropper. More specifically, the power beacons radiated the jamming signal during the data transmission phase to reduce the eavesdropper’s CSI. However, reference [20] did not consider cluster networks.

The authors in [21] considered multi-hop transmission under multiple eavesdroppers over Nakagami-*m* fading channel. However, the authors did not consider node selection to enhance secrecy performance in the considered model. The author in [22] studied the secrecy performance in multi-hop relaying systems, where the relays worked on full-duplex mode. However, this work had some limitations that the full-duplex relay could cancel the self-interference, and the author did not consider any technique to enhance the secrecy performance on the considered system similar to [21]. The authors in [23] exploited the secrecy performance in multi-hop transmission under the underlay cognitive radio networks. The authors addressed that the CSI of interference link is imperfect. Interestingly, in the case of one primary user, the imperfect CSI did not affect the secrecy performance. From the aforementioned works [21,22,23], the multi-hop transmission shows that the system secrecy performance decreases when the number of hops increases.

In order to enhance the secrecy performance in multi-hop transmission, some authors have addressed the opportunistic scheduling in multi-hop transmission systems. The authors in [24] proposed opportunistic scheduling scheme in cluster networks to improve the secrecy performance. The nodes were selected to improve the main channel performance. Additionally, the eavesdropper employed the MRC technique to enhance wiretapping performance. Recently, the authors in [25] addressed the path-selection method to enhance the secrecy performance in multi-hop transmission for wireless sensor networks. Additionally, the authors exploited the impact of hardware impairment in secrecy performance. In [26], the authors exploited the cooperative multi-hop transmission protocol in underlay cognitive radio networks. The authors considered hardware impairments for a more practical scenario. However, the authors did not propose any technique to improve the system secrecy performance. From the numerical results, in spite of node/path selection schemes, the considered system achieves very low secrecy performance. The PLS is addressed in multi-hop transmission in [23,24,25,26]. Thus, future research needs to consider not only opportunistic scheduling scheme but also some techniques to enhance the secrecy performance in multi-hop scenarios.

The aforementioned works [16,18,19,20] mainly focused on EH-based multi-hop transmission or [21,22,23,24,25,26] considered multi-hop secure transmission, which is not feasible in practice. In practical WSNs, because of the limited antenna performance of sensor nodes and deep shadowing, the direct link between source node and destination node is not available. The sensor nodes cooperate with other sensor nodes in order to overcome the channel attenuation between the source node and destination node. However, different from the legitimate users transmission, the eavesdropper overhears the legitimate users transmission in the vicinity. Thus, in the WSNs, sensor nodes are densely deployed and grouped into clusters as well as the existing wiretapping attacks. The study of PLS in WPT-based multi-hop transmissions is necessary for WSNs. Additionally, from [23,24,25,26], the multi-hop transmissions show low secrecy performance when the number of hops is increased. Thus, the PLS in WPT-based multi-hop transmissions plays a vital role in the future WSNs and IoT systems.

### 1.2. Contributions and Organization

In this paper, we study the effects of node selection on the performance of WPT-based multi-hop transmissions in WSNs. Specifically, we propose two node selection schemes to improve the secrecy performance, named the minimum node selection (MNS) scheme and optimal node selection (ONS) scheme. In particular, the MNS scheme selects the sensor node in each cluster based on the minimization of the eavesdropper channel capacity, while the ONS scheme tries to choose the best sensor node that maximizes the secrecy capacity in each hop transmission. Furthermore, the power beacons are deployed to not only charge the sensor nodes in energy harvesting phase but also generate the jamming signal to confuse the eavesdropper in data transmission phase. By employing the randomize-and-forward strategy [27,28], the considered cooperative relaying prevents the eavesdropper from combining the wiretapped information during the data transmission phase. The main contributions of this paper can be summarized as follows:We deploy an effective secure model for WPT-based multi-hop transmission to improve the energy capability of sensor nodes and reduce the decoding ability of eavesdropper in WSNs. Specifically, the dedicated RF signal for powering sensor nodes can be used to interfere the wiretapping channels. The considered WPT-based multi-hop transmission has been reported in this literature.We propose two scheduling schemes to improve the secrecy performance in WSNs. We then derive the new exact closed-form expressions for the SOP of the proposed schemes. The asymptotic analyses of the SOP are further provided to show some insight information into the considered system. The analytical results are verified by Monte-Carlo simulations confirming the correctness of our analysis.We show through the numerical results that the ONS scheme provides the best secrecy performance among the proposed schemes. Additionally, the effects of the number of sensor nodes, the number of PBs, and time switching ratio on the secrecy performance are also evaluated and discussed.

The rest of the paper is organized as follows. Section 2 describes the system model. Section 3 analyzes the exact closed-form expression for SOP of the proposed schemes. Section 4 presents the numerical results based on the derived analytical results. Finally, the paper is concluded in Section 5.

## 2. System Model

### 2.1. System Description and Channel Modeling

Let us consider a WPT-based multi-hop transmission in wireless sensor networks, as depicted in Figure 1, where a desired source in the first cluster transmits its information to a destination with the help of intermediate nodes located in *K*-1 cluster, with *K* > 1, while an eavesdropper overhears the signals transmitted by the source and relays. In this paper, we assume that sources and intermediate nodes have a limited power supply; thus, they must harvest energy from RF signal radiated by a set of *M* power beacons, i.e., P = {${\mathrm{PB}}_{m}$| *m* = 1, 2, ..., *M*}. We assume that the power beacons have the same structure and transmit the power at the same level, i.e., $P_{m}$ = *P* [19]. Moreover, all sensor nodes are equipped with a single antenna and operated in half-duplex mode. We further assume that the direct link from the source node to the destination node is not available due to the limited radio range of each sensor node and deep shadowing [19,25,29].

The main notations used in this paper are summarized in Table 1. Moreover, we assume that all channels exhibit flat and block Rayleigh fading. Let $u_{XY}$ be the channel coefficient from X to Y, where $X\in \{S_{i},R_{k,i}$, ${\mathrm{PB}}_{m}$} and Y∈ {$S_{i}$, $R_{k,i}$, D, E } with X≠Y. Thus, the channel gain $|u_{XY}{|}^{2}$ follows the exponential distribution whose cumulative distribution function (CDF), $F_{|u_{XY}{|}^{2}}\left(u\right)$ = 1 $-\exp(-\frac{u}{\lambda_{\mathrm{XY}}})$, and probability density function (PDF), $f_{|u_{XY}{|}^{2}}\left(u\right)$ = $\frac{1}{\lambda_{\mathrm{XY}}}\exp\left(-\frac{u}{\lambda_{\mathrm{XY}}}\right)$, respectively, where $\lambda_{\mathrm{XY}}$ indicates the mean of $|u_{\mathrm{XY}}{|}^{2}$ and can be expressed as $\lambda_{\mathrm{XY}}$ = ${\left(d_{\mathrm{XY}}/d_{0}\right)}^{-\epsilon }$, where $d_{\mathrm{XY}}$ presents the Euclidean distance between X and Y, $d_{0}$ and ϵ denote the reference distance and path-loss exponent, respectively.

The operation of the considered system can be divided into two consecutive phases including energy harvesting and data transmission which are presented in the next subsection. Additionally, the aims of the power beacons in each phase, called energy harvesting and data transmission phase, can be summarized as follows:Energy harvesting phase: According to wireless power transfer technique [9], the power beacons support the sensor nodes to harvest energy. Thus, in the energy harvesting phase, sensor nodes can harvest energy from the power beacons.Data transmission phase: In multi-hop transmission, sensor nodes transmit information by consuming the harvested energy. Thus, the eavesdropper can overhear the legitimate user’s information. In order to enhance the secrecy performance, the power beacons radiate the jamming signal to degrade the received signal-to-noise ratio (SNR) of the eavesdropper.

### 2.2. Energy Harvesting Phase

Figure 2 illustrates the time block structure of the considered system employing time switching architecture [9], where all sensor nodes simultaneously harvest energy from the *M*
PBs in duration of αT while the data transmissions is taken place in *K* orthogonal sub-time slots. The harvested energy at $R_{k,i}$ can be expressed as
(1)$$E_{k,i}=\sum_{m=1}^{M}\eta \alpha TP{\left|h_{k,m,i}\right|}^{2}.$$

Therefore, the average transmit power of $R_{k,i}$ in a sub-time slot can be calculated as
(2)$$\begin{array}{cc} P_{k,i}= & \frac{E_{k,i}}{(1-\alpha )T/K}\\ = & \sum_{m=1}^{M}\frac{K\eta \alpha P|h_{m,k,i}{|}^{2}}{1-\alpha }. \end{array}$$

### 2.3. Data Transmission Phase

In this phase, the source nodes transmit their data to the destination node via multiple intermediate sensor nodes while the power beacons generate the artificial noise to degrade the wiretapping channel of the eavesdropper. More specifically, as can be seen in Figure 2, *k*-th hop data transmission slot indicates that the transmission between a sensor node, called *i* node in (*k*-1)-th relay cluster, and an other sensor node, named *j* node in *k*-th relay cluster. Because each sensor node equips with half-duplex mode, in *k*-th time slot, the received sensor node in *k*-th relay cluster can transmit its message to an other sensor node in the next transmission slot. Simultaneously, the power beacon radiates the jamming signal as a role of friendly jammer in data transmission phase. We assume that each legitimate user and PBs cooperate such that the jamming signal can be nulled out at legitimate users [30,31]. Thus, the received signal at $R_{k+1,j}$ can be expressed as
(3)$$y_{k,i,j}=\sqrt{P_{k,i}}h_{k,i,j}x_{k,i}+n_{j},$$
where the $h_{k,i,j}$ denotes the channel coefficients of $R_{k,i}\to R_{k+1,j}$ link and $n_{j}$ presents the additive white Gaussian noise (AWGN) at $R_{k+1,j}$ with zero mean and variance $\sigma_{R_{k+1,j}}^{2}$. Without loss of generality, we assume that the each node has the same variance of noise, i.e., $\sigma_{R_{k+1,j}}^{2}$ = $\sigma^{2}$. The instantaneous SNR of the main channel at *k* hop can be expressed as
(4)$$\gamma_{k,i,j}=\frac{P_{k,i}{\left|h_{k,i,j}\right|}^{2}}{\sigma^{2}}.$$

Meanwhile, the eavesdropper can overhear the information from the legitimate users due to the nature broadcast of the wireless signals. Thus, the wiretapped signals in *k*-th hop transmission at E can be expressed as
(5)$$y_{k,i,E}=\sqrt{P_{k,i}}h_{k,i,E}x_{k,i}+\sum_{m=1}^{M}\sqrt{P}h_{k,m,E}s_{m}+n_{E}.$$

Since the power beacons generate the jamming signals to degrade the decoding ability of the eavesdropper. Thus, the instantaneous signal-to-interference-plus-noise ratio (SINR) of *k*-th hop at E can be expressed as
(6)$$\gamma_{k,i,E}=\frac{P_{k,i}{\left|h_{k,i,E}\right|}^{2}}{\sum_{m=1}^{M}P{\left|h_{k,m,E}\right|}^{2}+\sigma^{2}}.$$

In order to facilitate the analysis and highlight the secure performance under different opportunistic scheduling schemes, we assume that the perfect CSI is available at all receivers [24,25,26,32,33] (In [34], the authors proposed the method of the eavesdropper channel estimation without eavesdropper feedback. The other legitimate users, called torch nodes, feedback their channel information instead of the eavesdropper. Since we mainly focus on the impact of the node selection schemes in the considered SOP performance of WPT-based multi-hop transmission. The study of imperfect CSI of eavesdropper link in our system model is indeed an interesting topic and will be considered in our future works).

### 2.4. Opportunistic Scheduling Scheme

In this subsection, we explain the proposed opportunistic scheduling schemes for multi-hop transmission in WSNs. In order to analyze the impacts of scheduling schemes on the secure performance, we assume that various techniques, e.g., clustering protocols [35,36], eavesdropper channel estimation [34], distributed coordination function (DCF) in IEEE 802.11 [37,38], etc., perfectly support the proposed node selection schemes as in [24,26,32,33].

#### 2.4.1. Random Node Selection Scheme

We consider the random node selection scheme as a baseline scheme for comparison purpose. In particular, the RNS scheme randomly selects a relay in each cluster for multi-hop transmission [19]. Thus, the instantaneous SNRs of main channel and eavesdropper channel can be expressed, respectively, as
(7)$$\gamma_{k,i^{*},j^{*}}^{\mathrm{RNS}}=\frac{P_{k,i}{\left|h_{k,i,j}\right|}^{2}}{\sigma^{2}},$$
(8)$$\gamma_{k,i^{*},E}^{\mathrm{RNS}}=\frac{P_{k,i}{\left|h_{k,i,E}\right|}^{2}}{\sum_{m=1}^{M}P{\left|h_{k,m,E}\right|}^{2}+\sigma^{2}}.$$

#### 2.4.2. Minimum Node Selection Scheme

In MNS scheme, the selected node is the least vulnerable node among the nodes in each cluster, which minimizes the channel capacity of eavesdropper channel in each cluster. Thus, MNS scheme can be mathematically described as
(9)$$R_{k,i^{*}}^{\mathrm{MNS}}=\arg\min_{i\in N_{k}}\left\{\log_{2}\left(1+\gamma_{k,i,E}\right)\right\}.$$

The instantaneous SNRs of main channel and eavesdropper channel can be expressed, respectively, as
(10)$$\gamma_{k,i^{*},j^{*}}^{\mathrm{MNS}}=\frac{P_{k,i^{*}}{\left|h_{k,i^{*},j^{*}}\right|}^{2}}{\sigma^{2}},$$
(11)$$\gamma_{k,i^{*},E}^{\mathrm{MNS}}=\arg\min_{i\in N_{k}}\left\{\frac{P_{k,i}{\left|h_{k,i,E}\right|}^{2}}{\sum_{m=1}^{M}P{\left|h_{k,m,E}\right|}^{2}+\sigma^{2}}\right\}.$$
where $P_{k,i^{*}}$ and $|h_{k,i^{*},j^{*}}{|}^{2}$ indicate the average transmit power and channel gain of the selected sensor node in *k*-th cluster through the MNS scheme, respectively.

#### 2.4.3. Optimal Node Selection Scheme

In this scheme, the main and eavesdropper channels are jointly considered, thus, ONS scheme can achieve the most robust performance [33,39]. Since the ONS scheme utilizes both the main and eavesdropper channel to enhance secrecy performance, the selection criteria can be mathematically described as
(12)$$R_{k,i^{*}}^{\mathrm{ONS}}=\arg\max_{i\in N_{k}}\left\{\log_{2}\left(\frac{1+\gamma_{k,i,j^{*}}}{1+\gamma_{k,i,E}}\right)\right\},$$
where $j^{*}$ indicates the selected node which has already chosen in the previous hop.

## 3. Outage Performance Analysis

In this section, we analyze the effects of each scheduling scheme on the secrecy performance of the considered system setup. We evaluate the secrecy outage probability which is defined as [23,33,40]
(13)$$P_{out}^{\mathrm{sch}}=\mathrm{Pr}\left(\frac{1-\alpha }{K}\min_{k\in K}\log_{2}\left(\frac{1+\gamma_{k,i^{*},j^{*}}^{\mathrm{sch}}}{1+\gamma_{k,i^{*},E}^{\mathrm{sch}}}\right)<R_{th}^{\mathrm{sch}}\right),$$
where sch∈ {RNS, MNS, ONS} and $R_{th}^{\mathrm{sch}}$ (bps/Hz) indicate the secrecy target data rate of each node selection scheme. For the sake of notational convenience, κ≜$\frac{K\eta \alpha }{\left(1-\alpha \right)}$, γ≜$P/\sigma^{2}$, $X_{k,m,i}$≜$|h_{k,m,i}{|}^{2}$, $Y_{k,i,j}$≜$|h_{k,i,j}{|}^{2}$, $Z_{k,i,E}$≜$|h_{k,i,E}{|}^{2}$, $T_{k,m,E}$≜$|h_{k,m,E}{|}^{2}$, $W_{k,i}$≜$\sum_{m=1}^{M}X_{k,m,i}$, and $V_{k,E}$≜$\sum_{m=1}^{M}T_{k,m,E}$, respectively.

### 3.1. Exact Closed-Form Expression for SOP Analysis

#### 3.1.1. RNS Scheme

**Theorem** **1.**
*The exact closed-form expression for the SOP of RNS scheme can be derived as*
(14)$$\begin{array}{cc} P_{\mathrm{out}}^{\mathrm{RNS}}=1-\prod_{k=1}^{K}[ & {\left(\frac{1}{\lambda_{k,m,i}\lambda_{k,m,E}}\right)}^{M}\frac{2}{\Gamma (M)\Gamma (M)}{\left(\frac{\lambda_{k,m,i}\left(\gamma_{th}^{\mathrm{RNS}}-1\right)}{\kappa \gamma \lambda_{k,i,j}}\right)}^{M/2}K_{M}\left(2\sqrt{\frac{\left(\gamma_{th}^{\mathrm{RNS}}-1\right)}{\kappa \gamma \lambda_{k,m,i}\lambda_{k,i,j}}}\right)\\ \; & \times \left(\Gamma \left(M\right)\lambda_{k,m,E}^{M}-\frac{\gamma_{th}^{\mathrm{RNS}}\lambda_{k,i,E}}{\gamma \lambda_{k,i,j}}\beta_{1}^{M-1}\exp\left(\frac{\beta_{1}}{\lambda_{k,m,E}}\right)\Gamma \left(M\right)\Gamma \left(1-M,\frac{\beta_{1}}{\lambda_{k,m,E}}\right)\right)], \end{array}$$
*where $\gamma_{th}^{\mathrm{RNS}}≜2^{\frac{KR_{th}^{\mathrm{RNS}}}{\left(1-\alpha \right)}}$, $\beta_{1}$ = ($\gamma_{th}^{\mathrm{RNS}}\lambda_{k,i,E}+\lambda_{k,i,j}$)/ $\gamma \lambda_{k,i,j}$, $K_{\nu }\left(\cdot \right)$ is the modified Bessel function of the second kind with order ν ([41], eq. 8.342.6), Γ(z) is the Gamma function ([41], 8.310.1), and Γ(α,x) is the upper incomplete Gamma function ([41], eq. 8.350.2), respectively.*


**Proof.** See Appendix A. □

#### 3.1.2. MNS Scheme

**Theorem** **2.**
*The exact closed-form expression for the SOP of MNS scheme can be derived as*
(15)$$\begin{array}{cc} P_{out}^{\mathrm{MNS}}= & 1-\prod_{k=1}^{K}[{\left(\frac{1}{\lambda_{k,m,i}\lambda_{k,m,E}}\right)}^{M}\frac{2}{\Gamma (M)\Gamma (M)}{\left(\frac{\left(\gamma_{th}^{\mathrm{MNS}}-1\right)\lambda_{k,m,i}}{\kappa \gamma \lambda_{k,i,j}}\right)}^{M/2}K_{M}\left(2\sqrt{\frac{\gamma_{th}^{\mathrm{MNS}}-1}{\kappa \gamma \lambda_{k,m,i}\lambda_{k,i,j}}}\right)\\ & \times \left[\Gamma \left(M\right)\lambda_{k,m,E}^{M}-\frac{\gamma_{th}^{\mathrm{MNS}}\lambda_{k,i,E}}{\gamma N\lambda_{k,i,j}}\beta_{2}^{M-1}\exp\left(\frac{\beta_{2}}{\lambda_{k,m,E}}\right)\Gamma \left(M\right)\Gamma \left(1-M,\frac{\beta_{2}}{\lambda_{k,m,E}}\right)\right]], \end{array}$$
*where $\gamma_{th}^{\mathrm{MNS}}≜2^{\frac{KR_{th}^{\mathrm{MNS}}}{\left(1-\alpha \right)}}$ and $\beta_{2}$ = ( $\gamma_{th}^{\mathrm{MNS}}\lambda_{k,i,E}+N\lambda_{k,i,j}$ )/ $\gamma N\lambda_{k,i,j}$.*


**Proof.** See Appendix B. □

#### 3.1.3. ONS Scheme

**Theorem** **3.**
*The exact closed-form expression for the SOP of ONS scheme can be derived as*
(16)PoutONS=1−∏k=1K[1−1λk,m,EM1Γ(M)∑n=0N∑i=0nNnni(−1)n+iγthONSλk,i,E+λk,i,jγλk,i,ji×1λk,m,iM2Γ(M)(γthMNS−1)λk,m,iκγλk,i,jM/2KM2γthMNS−1κγλk,m,iλk,i,jn×expβ3λk,m,E1λk,m,E−M−i+12β3M−i−12Γ(M)exp−β32λk,m,EW−i+1−M2,i−M2β3λk,m,E],
*where $\gamma_{th}^{\mathrm{ONS}}≜2^{\frac{KR_{th}^{\mathrm{ONS}}}{\left(1-\alpha \right)}}$, $\beta_{3}$ = $\left(\gamma_{th}^{\mathrm{ONS}}\lambda_{k,i,E}+\lambda_{k,i,j}\right)$/$\gamma \lambda_{k,i,j}$ and $W_{\lambda ,\mu }\left(z\right)$ presents the Whittaker function ([41], eq. 9.220.4).*


**Proof.** See Appendix C. □

### 3.2. Asymptotic SOP Analysis

In this subsection, we consider the asymptotic expressions of SOP with both schemes in order to insights when $P_{m}$→*∞*. In order to obtain asymptotic SOP, we apply the following approximation for small $z_{k}$ as [18]
(17)$$\begin{array}{c} \prod_{k=1}^{K}\left(1-z_{k}\right)\approx 1-\sum_{k=1}^{K}z_{k}. \end{array}$$

#### 3.2.1. RNS Scheme

**Corollary** **1.**
*The asymptotic SOP of RNS scheme can be derived as*
(18)$$P_{Asym}^{\mathrm{RNS}}=\sum_{k=1}^{K}\Delta_{asym},$$
*where*
(19)$$\begin{array}{cc} \Delta_{asym}=1- & {\left(\frac{1}{\lambda_{k,m,E}\lambda_{m,k,i}}\right)}^{M}\frac{2}{\Gamma (M)}{\left(\frac{\left(\gamma_{th}^{\mathrm{RNS}}-1\right)\lambda_{m,k,i}}{\kappa \gamma \lambda_{k,i,j}}\right)}^{M/2}K_{M}\left(2\sqrt{\frac{\left(\gamma_{th}^{\mathrm{RNS}}-1\right)}{\kappa \gamma \lambda_{k,m,i}\lambda_{k,i,j}}}\right)\\ \; & \times \left[\lambda_{k,m,E}^{M}-\frac{\gamma_{th}^{\mathrm{RNS}}\lambda_{k,i,E}}{\gamma \lambda_{k,i,j}}\beta_{1}^{M-1}\exp\left(\frac{\beta_{1}}{\lambda_{k,m,E}}\right)\Gamma \left(1-M,\frac{\beta_{1}}{\lambda_{k,m,E}}\right)\right]. \end{array}$$


**Proof.** From (Equation 14), we apply (Equation 17), $P_{Asym}^{\mathrm{RNS}}$ can be further written as
(20)$$\begin{array}{cc} P_{Asym}^{\mathrm{RNS}}= & 1-\left[1-\sum_{k=1}^{K}\Delta_{asym}\right]\\ = & \sum_{k=1}^{K}\Delta_{asym}, \end{array}$$
where $\Delta_{asym}$ is defined as in (Equation 19). The proof of Corollary 1 is concluded. □

#### 3.2.2. MNS Scheme

**Corollary** **2.**
*The asymptotic SOP of MNS scheme can be derived as*
(21)$$\begin{array}{c} P_{Asym}^{\mathrm{MNS}}=\sum_{k=1}^{K}\Phi_{asymn}, \end{array}$$
*where the following notation is adopted*
(22)$$\begin{array}{cc} \Phi_{asym}=1- & {\left(\frac{1}{\lambda_{k,m,i}\lambda_{k,m,E}}\right)}^{M}\frac{2}{\Gamma (M)}{\left(\frac{\left(\gamma_{th}^{\mathrm{MNS}}-1\right)\lambda_{k,m,i}}{\kappa \gamma \lambda_{k,i,j}}\right)}^{M/2}K_{M}\left(2\sqrt{\frac{\gamma_{th}^{\mathrm{MNS}}-1}{\kappa \gamma \lambda_{k,m,i}\lambda_{k,i,j}}}\right)\\ \; & \times \left[{\left(\frac{1}{\lambda_{k,m,E}}\right)}^{-M}-\frac{\gamma_{th}^{\mathrm{MNS}}\lambda_{k,i,E}}{\gamma N\lambda_{k,i,j}}\beta_{2}^{M-1}\exp\left(\frac{\beta_{2}}{\lambda_{k,m,E}}\right)\Gamma \left(1-M,\frac{\beta_{2}}{\lambda_{k,m,E}}\right)\right]. \end{array}$$


**Proof.** Similar to Corollary 1, the asymptotic SOP of MNS scheme can easily obtained as (Equation 21). The proof of Corollary 2 is concluded. □

#### 3.2.3. ONS Scheme

**Corollary** **3.**
*The asymptotic SOP of ONS scheme can be derived as*
(23)$$P_{Asym}^{\mathrm{ONS}}=\sum_{k=1}^{K}\Psi ,$$
*where *Ψ* is defined as in (Equation 73).*


**Proof.** Similar to Corollary 1, the asymptotic SOP of ONS scheme can easily obtained as (Equation 23). The proof of Corollary 3 is concluded. □

## 4. Numerical Results

In this section, we present representative numerical results to illustrate the achieve secrecy outage performance of the proposed schemes. Unless otherwise stated, the simulation parameters are presented in Table 2 [19,20].

First of all, we exploit the impact of the energy parameters, i.e., transmit SNR of power beacons, the number of power beacons, and time switching ratio, to the average harvested energy of sensor node cluster. As can be seen in Figure 3, the average harvested energy of the sensor node cluster increases when the energy parameters is increased. In detail, in Figure 3a, the average harvested energy is linearly increased when transmit SNR of power beacons increases. Different from Figure 3a, when the number of power beacons increases, the slope of the harvested energy of sensor node cluster is decreased as shown in Figure 3b. The slope of the average harvested energy of sensor node cluster is decreased when the time switching ratio increases as shown in Figure 3c. From Figure 3, when transmit SNR of power beacons, the number of power beacons and time switching ratio increases, the sensor node can harvest enough energy to operate the sensor node [42]. It is noted that, as can be seen Table 2, the sensor node cluster is located at the most farthest from the power beacons. Thus, other clusters can naturally harvest more energy from the power beacons to transmit message and sense. Additionally, during the networks planning step, the engineers or administrators can make a network that the sensor nodes can sufficiently harvest the energy to transmit pilot and data messages or other source consumption.

We turn our attention to the impact of the system parameters on the secrecy outage performance. We studied the impact of transmit SNR, γ, on the SOP of the proposed schemes. As can be observed in Figure 4, the secrecy performances of all schemes were enhanced when the transmit SNR was increased. The reason is that the jamming signals significantly interfere with the eavesdropper channel when the transmit SNR of the power beacons is large. Additionally, the ONS scheme shows the robust performance among the proposed schemes since the ONS scheme required both CSI of the main and eavesdropper channels as in (Equation 12). In Figure 4, when the transmit SNR of power beacons was increased, the secrecy outage performance of the considered system model was significantly increased. One of possible reasons is that, in data transmission phase, the power beacons radiate the jamming signal to degrade the received SINR of the eavesdropper links.

Figure 5 illustrates the effect of the time switching ratio, α, on the SOP of the proposed schemes. The pattern of SOP presents the convex function that means the proposed scheme makes an adequate time switching ratio to improve system secrecy performance. When the time switching ratio was higher than 0.8, the system performance seemed to be saturated. Finally, the appropriately selected time switching ratio played an important role in network planning to enhance the overall system performance.

The effect of the secrecy target data rate, $R_{th}$, on the SOP is presented in Figure 6. A high secrecy target data means that the system requires a high system secrecy level. Thus, the outage event frequently occurs when system requires a high threshold. Furthermore, the SOPs of all schemes are increased when the secrecy target data rate increases. Different from the RNS and MNS schemes, the ONS scheme significantly enhances the system performance under the same channel settings. The reason is that the ONS scheme requires both main and eavesdropper channel information to select the node in each cluster to perform multi-hop transmission. Moreover, the results from Figure 4 to Figure 6 show that the theoretical results are in good agreement with the simulation ones validating the correctness of our derivation approaches.

Now, we turn our attention to the effects of the number of hops, number of nodes in each cluster and number of power beacons on the system secrecy performance. Figure 7 plots the SOPs as a function of the number of hops. As can be observed, ONS scheme showed better secrecy performance than that of MNS and RNS schemes. The reason is that the RNS scheme does not consider the channel information to select the sensor node in each hop while MNS scheme only considers the eavesdropper channel. Differently, ONS scheme considers both main and eavesdropper channel information to select the best sensor node in each hop to perform the multi-hop transmission. Thus, the secrecy performance of ONS scheme outperforms the other proposed schemes in the same system setup. Similar to the time switching ratio, the number of hops shows the convex patterns in all node selection schemes; thus, appropriately selecting the number of hops plays an important role in network planning to enhance the system secrecy performance.

Figure 8 presents the SOPs as a function of the number of sensor node in each cluster. As can be observed, when the number of nodes in each cluster was increased, the SOPs of ONS and MNS schemes were increased, while the SOP of RNS scheme was still unchanged. The reason is that RNS scheme randomly selects a node in each cluster for multi-hop transmission. More specifically, the MNS scheme requires the eavesdropper channel information to select the sensor node in each hop. Thus, the SOP of MNS scheme shows better performance than that of RNS scheme. Different from other schemes, since ONS scheme utilizes both main and eavesdropper channel information to select the node, ONS scheme dramatically enhances the secrecy performance compared to that of the other schemes when the number of node in each cluster increases.

Figure 9 illustrates the SOP as a function of the number of power beacons. The SOPs of the proposed schemes were enhanced when the number of power beacons was increased. The reason is that the SINR of eavesdropper channel degrades when the number of power beacons increases in (Equation 6). As can be observed, the SOPs of MNS and ONS schemes are significantly improved compared to that of RNS scheme. The reason is that MNS scheme consider the CSI of eavesdropper channel to select the sensor node in a each cluster while the ONS scheme utilizes both the CSI of main channel and eavesdropper channels to select the best sensor node in a each cluster. Thus, the secrecy performance is significantly enhanced under the same system setup. In Figure 8 and Figure 9, when the number of sensor nodes in each cluster and the number of power beacons is increased, the system secrecy performance is increased excepted the RNS scheme. It is noted that increasing the number of sensor nodes in each cluster significantly enhances the secrecy performance compared to that of increasing the number of power beacons in the same network settings.

Finally, we exploit the complexity order of each node selection scheme as in Table 3. Complexity order represents the required channel estimation to select the node and transmit information [43,44]. As can be seen in Table 3, the amount of channel information of the RNS scheme was the smallest among the proposed scheduling schemes. The reason is that the RNS scheme does not require the channel information to select the node. Thus, the total required channel information is *K*. Differently, MNS scheme needs to estimate the channel information to select the best node. Thus, the total complexity order is $(M+N_{k}+1)K$. Optimally, ONS scheme utilizes both main and eavesdropper channels information to select the best node. Since the nodes are sequentially selected from the destination node, the amount of the required channel information at a certain hop transmission is $2N_{k}+MN_{k}+M$, and the amount of the required channel information for *K* hops transmission is $(2N_{k}+MN_{k}+M)K$. Through Table 3, each scheme has a specific advantage and drawback. Thus, in network planning perspective, each scheme can be cleverly applied in the practice to achieve a good trade-off between secrecy performance and complexity.

## 5. Conclusions

This paper proposed the system model and the scheduling scheme to enhance the secrecy performance for WPT-based multi-hop transmission in WSNs. More specifically, the power beacon served as a friendly jammer to reduce the CSI of eavesdropper as well as the radiation energy to harvest the sensor node. We proposed two kinds of node selection schemes, called as MNS and ONS schemes, to improve the secrecy performance. The MNS scheme selected the sensor node to minimize the eavesdropper channel information in each cluster while ONS schemes utilized both main and eavesdropper channel information to select the best cooperative node in each cluster. We derived the exact closed-form expression for SOP of the proposed schemes. In addition, to provide more insights into the proposed schemes, we derived asymptotic SOP. From the numerical results, the secrecy performance of the ONS scheme outperformed compared to that of MNS scheme under the same system setup. However, through the complexity order analysis, the ONS scheme showed the more required channel information to select the sensor node and transmit information than that of MNS scheme.

## Figures and Tables

**Figure 1 sensors-19-05456-f001:**
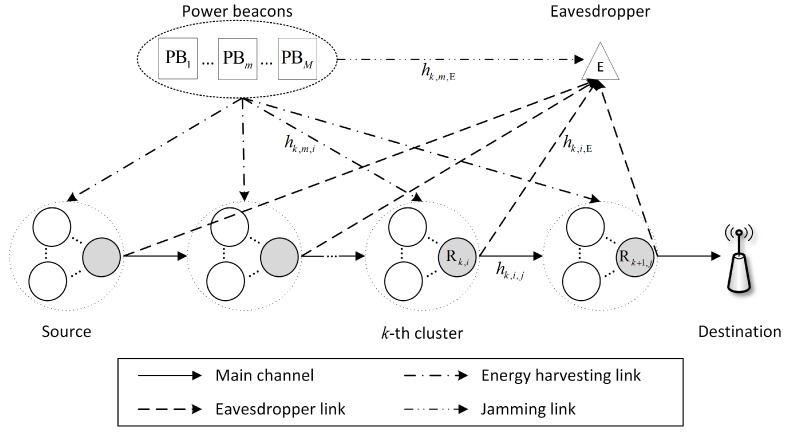
Illustration of the proposed multi-hop transmissions in wireless sensor network.

**Figure 2 sensors-19-05456-f002:**
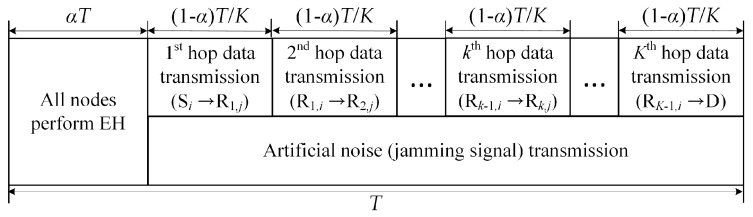
The transmission time block structure of the wireless power transfer (WPT)-based multi-hop transmissions in wireless sensor network for energy harvesting and data transmission.

**Figure 3 sensors-19-05456-f003:**
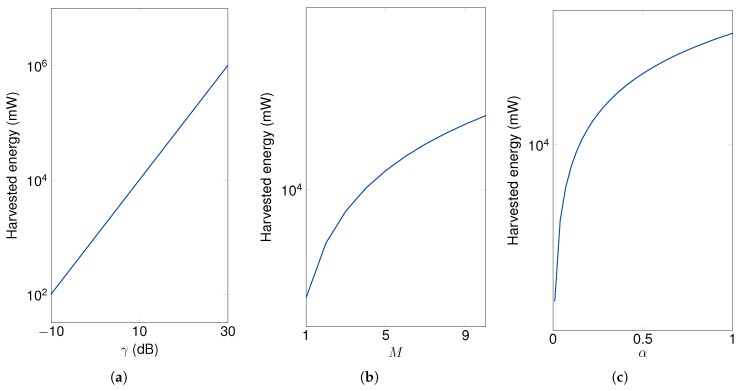
The impact of energy parameters on the average harvested energy of the cluster of sensor nodes with *M* = 4, $N_{k}$ = 2, *K* = 3, η = 0.7, α = 0.15, γ = 10 dB and $R_{th}$ = 1 bps/Hz. (**a**) Energy harvesting (EH) versus γ; (**b**) EH versus *M*; (**c**) EH versus α.

**Figure 4 sensors-19-05456-f004:**
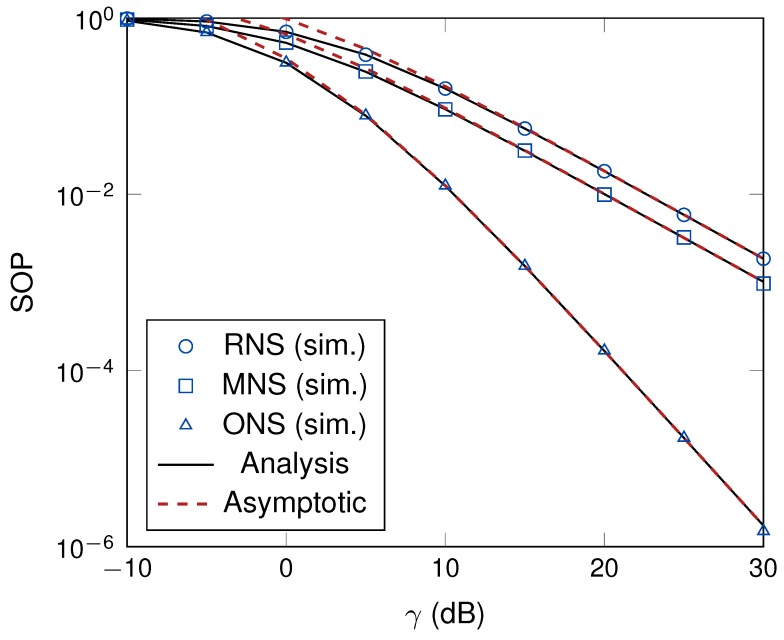
The illustration of the effect of γ on the secrecy outage probability (SOP) with *M* = 4 and *K* = 3.

**Figure 5 sensors-19-05456-f005:**
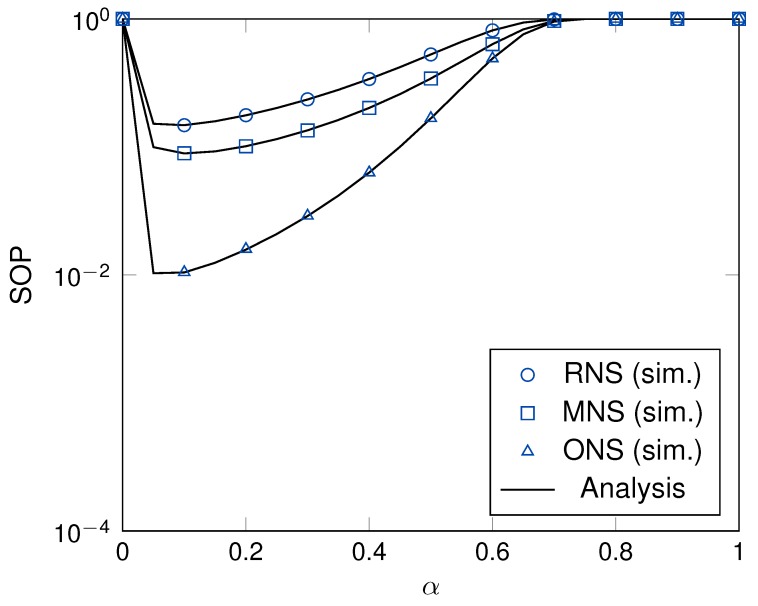
The SOP versus α with γ = 10 dB, *M* = 4 and *K* = 3.

**Figure 6 sensors-19-05456-f006:**
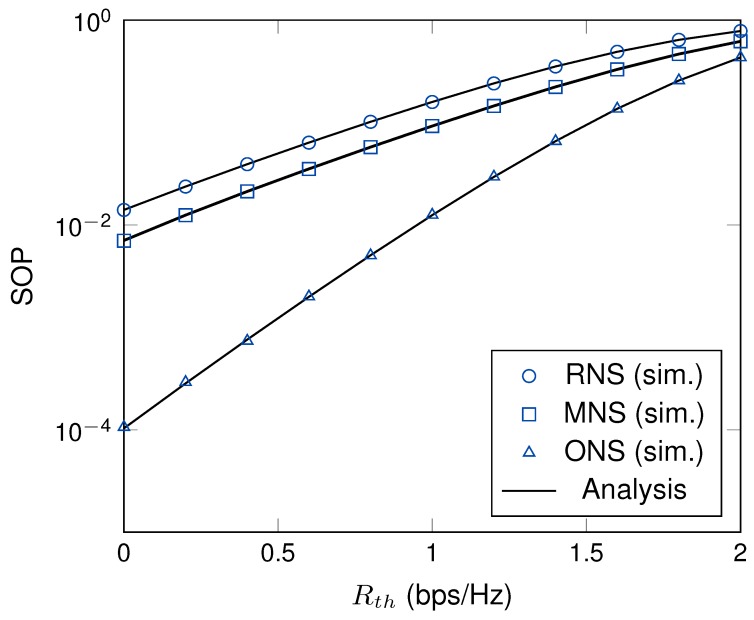
The effect of the SOP versus $R_{th}$ with γ = 10 dB, *M* = 4, *K* = 3.

**Figure 7 sensors-19-05456-f007:**
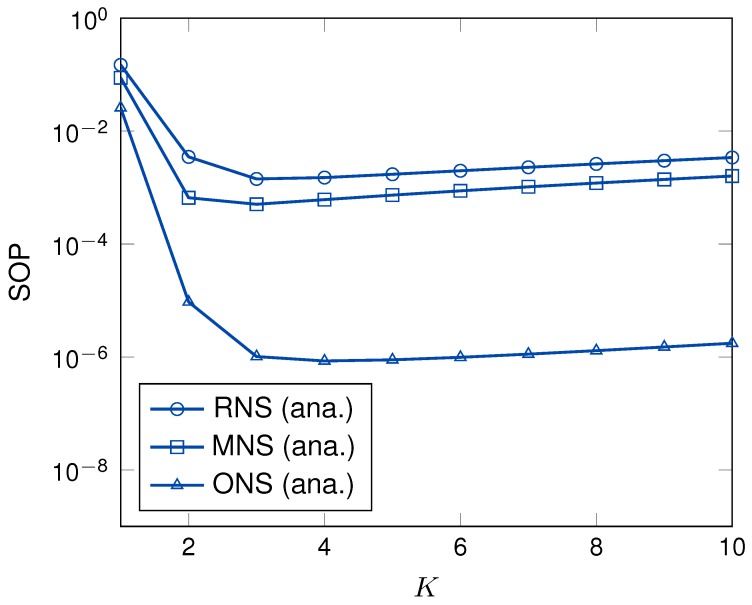
The SOP versus *K* with γ = 5 dB, $R_{th}$ = 0.1 bps/Hz, *M* = 4.

**Figure 8 sensors-19-05456-f008:**
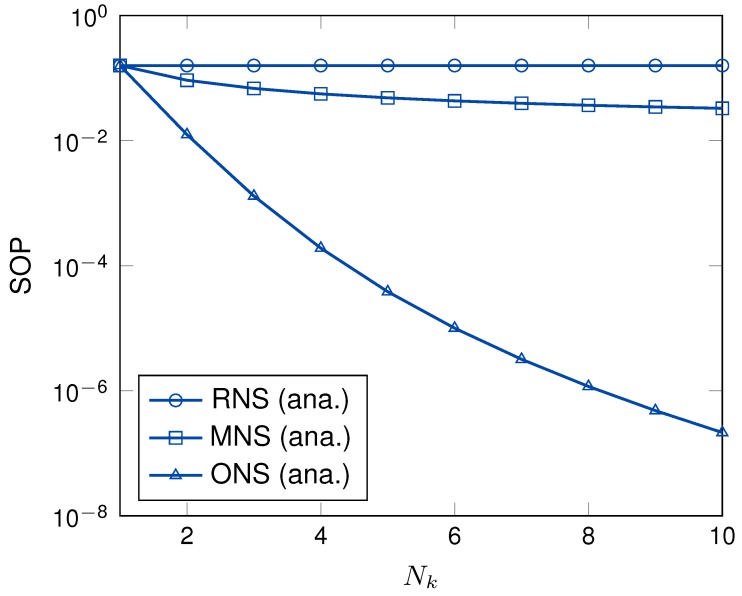
The SOP versus $N_{k}$ with γ = 10 dB, *M* = 4, *K* = 3.

**Figure 9 sensors-19-05456-f009:**
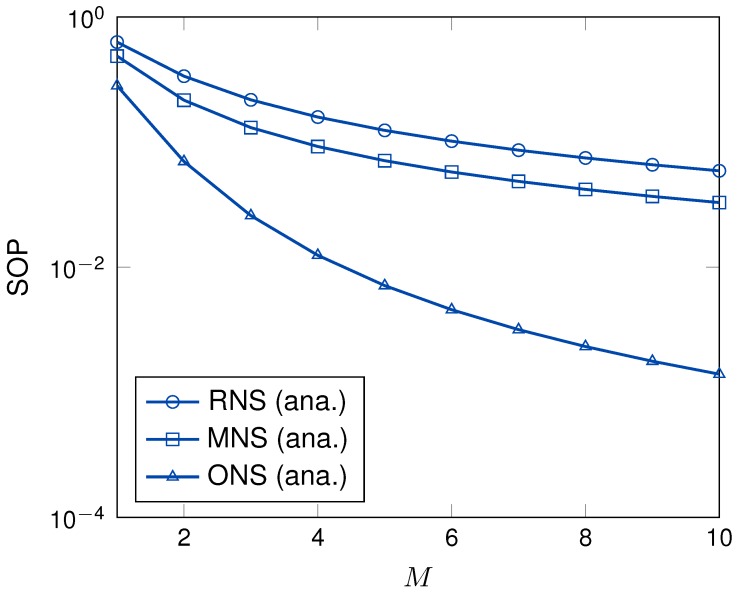
The SOP versus *M* with γ = 15 dB, *K* = 3.

**Table 1 sensors-19-05456-t001:** Summary of main notations.

Symbol	Definition
${\mathrm{PB}}_{m}$	Power beacons *m* with *m* ∈ {1, 2, ..., *M*}.
$S_{i}$	Source node in the first cluster, where *i* ∈ {1, ...,$N_{0}$}.
D	Destination node.
$N_{k}$	The number of relay in the *k*-th cluster, *k* ∈ {0,1,..., *K*-1} and $N_{k}$ ≥ 1.
$R_{k,i}$	The *i*-th sensor node in the *k*-th cluster., $R_{0,i}$ ≡ $S_{i}$ and $R_{K,1}$ ≡ D with *k* ∈ {0, ..., *K* }.
E	The eavesdropper node.
$|h_{k,m,i}{|}^{2}$	The channel gain of link ${\mathrm{PB}}_{m}$ → $R_{k-1,i}$.
$|h_{k,m,E}{|}^{2}$	The channel gain of link ${\mathrm{PB}}_{m}$ → E.
$|h_{k,i,j}{|}^{2}$	The channel gain of link $R_{k-1,i}$ → $R_{k,j}$.
$|h_{k,i,E}{|}^{2}$	The channel gain of link $R_{k-1,i}$ → E.
$\lambda_{k,m,i}$, $\lambda_{k,i,j}$, $\lambda_{k,i,E}$ and $\lambda_{k,m,E}$	Mean of $|h_{k,m,i}{|}^{2}$, $|h_{k,i,j}{|}^{2}$, $|h_{k,i,E}{|}^{2}$ and $|h_{k,m,E}{|}^{2}$, respectively.
α with α∈[0,1]	Time switching ratio.
η with η∈(0,1)	The energy conversion efficiency.

**Table 2 sensors-19-05456-t002:** Summary of simulation parameters.

Parameters	Value
The distance between S and D, $d_{\mathrm{SD}}$	10 m
The reference distance, $d_{0}$	1 m
The position of S	(0, 0)
The position of $R_{k}$	($d_{\mathrm{SD}}$ *k*/*K*, 0)
The position of D	(10, 0)
The position of ${\mathrm{PB}}_{m}$	(7.5, 5.5)
The position of E	(−5, 5)
The number of relays in each cluster, $N_{k}$	2
The secrecy target data rate, $R_{th}^{\mathrm{sch}}$	1 bps/Hz
Pathloss exponent, β	2.7
Pathloss at reference distance, L at $d_{0}$	−30 dB
Energy conversion efficiency, η	0.7
Time switching ratio, α	0.15

**Table 3 sensors-19-05456-t003:** The comparison of complexity order in the proposed schemes.

Scheme	RNS	MNS	ONS
Complexity Order	*K*	$(M+N_{k}+1)K$	$(2N_{k}+MN_{k}+M)K$

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
