# Peer review of "Exploiting Opportunistic Scheduling Schemes and WPT-Based Multi-Hop Transmissions to Improve Physical Layer Security in Wireless Sensor Networks†"

_sensors, 2019, doi:10.3390/s19245456_

Round 1

Reviewer 1 Report

The authors have addressed all the comments in the previous round of the review process.

Reviewer 2 Report

The authors satisfactorily answered all the comments I made in the previous review.

Reviewer 3 Report

The paper is improved from the previous version.

I still think that assuming a perfect CSI knowledge (especially the CSI of the eavesdropper) is a quite unrealistic assumption. About the proposed method (torch nodes) a natural objection is that if the eavesdropper location is known, then it's far easier to physically remove the eavesdropper rather than placing torch nodes around it.

Nevertheless, even an unrealistic assumption can be useful to draw upper bound performance, and the authors seems to have done a good job on replying to the previous comments.

Reviewer 4 Report

The authors addressed properly the comments and the paper can be considered for publication

This manuscript is a resubmission of an earlier submission. The following is a list of the peer review reports and author responses from that submission.

Round 1

Reviewer 1 Report

Two node selection schemes are proposed to improve the secrecy performance of WPT-based multi-hop transmissions in WSNs. The work is based on scenarios and previous results presented by the same authors in [17] (WPT-based multi-hop transmission model in the presence of an eavesdropper, but without considering clustered relays), [20] (multi-hop transmission system in an underlay cognitive radio network) and [30] (a multirelay cooperative networks with a source relay selection scheme in the presence of an eavesdropper). In fact, the system model now being presented is basically the same as the one presented in [30] with the inclusion o power beacons for energy harvesting and to jam the eavesdropper as already used in [17]. For instance, the Minimum Node Selection (MNS) scheme being presented as one of the contributions of this work is similar to the PSRS scheme, already presented in [30]. While this application is interesting, there are some major issues to be addressed. Below are some comments:

- It was not specified which relay protocol was considered. From the mathematical development presented, this reviewer believes it is the decode-and-forward protocol. In this regard, and given that relays are half-duplex, each transmission would be expected to take place in two time slots. But this is not what is shown in the transmission block structure of Figure 2 and therefore it is not clear how this would affect the system performance in terms of outage probability.

- The authors assume that 'sources and intermediate nodes have a limited power supply; thus, they must harvest energy from RF signal radiated by a set of M power beacons'. They also assume that 'perfect CSI is available at all the receivers', without indicating how this would be achieved in practice. However, the analysis under perfect CSI could be misleading for a wireless-powered communication network, due to its inherent energy constraints. What happens if the harvested energy is insufficient for pilot transmission, information transmission and other consumption sources? A quantitative analysis regarding the energy harvesting process and its influence on system performance should be presented.

- The outage performance analysis is presented only in terms of the mutual information in the different links of the system. However, notice that in the presented energy-limited scenario, an outage event can happen for two reasons: when the transmitter has not the sufficient power to feed its circuits and send the estimating pilot or when there is data transmission but the receiver is unable to successfully decode the received message. The authors consider only the second reason. Quantitative analysis involving the power beacons is only considered to improve the degradation performance they provide in the eavesdropper channel.

Reviewer 2 Report

In this work, the authors investigated the secrecy outage performance of WPT-based multi-hop transmissions in WSNs. The topic is interesting, and the results seemed correct. After going through the paper, the reviewer believes this work is not suitable for publication in the current form. I recommend major revision for this work according to the following comments: 1. The authors miss the related works about the physical layer security of WPT systems, such as R1-R3, The authors need to carefully review the literature and mention the differences in their work compared to the available research. R1 "On secure underlay MIMO cognitive radio networks with energy harvesting and transmit antenna selection," IEEE Trans. Green Commun. Netw., Jun. 2017. R2 "Performance analysis and optimization for SWIPT wireless sensor networks," IEEE Trans. Commun., May 2017. R3 "3-D hybrid VLC-RF indoor IoT systems with light energy harvesting," IEEE Trans. Green Commun. Netw., Sept. 2019. 2. What's the aim of power beacons in the considered system? Why the eavesdropper can receive the signals from sensor nodes, while the destination must use the multi-hop? 3. More insights should be given to enhance the contribution of this work. For example, how the parameters influence the secrecy performance? What conclusions can be found from the results of this work? The authors should know that rigorous mathematical derivation cannot be a novel contribution.